# Rendering Immersive Haptic Force Feedback via Neuromuscular Electrical Stimulation

**DOI:** 10.3390/s22145069

**Published:** 2022-07-06

**Authors:** Elisa Galofaro, Erika D’Antonio, Nicola Lotti, Lorenzo Masia

**Affiliations:** Assistive Robotics and Interactive Exosuits (ARIES) Laboratory, Institute of Computer Engineering (ZITI), Heidelberg University, 69120 Heidelberg, Germany; erika.dantonio@ziti.uni-heidelberg.de (E.D.); nicola.lotti@ziti.uni-heidelberg.de (N.L.); lorenzo.masia@ziti.uni-heidelberg.de (L.M.)

**Keywords:** haptics, NMES, wearable device, virtual reality, immersive feedback, metabolic consumption, kinematics

## Abstract

Haptic feedback is the sensory modality to enhance the so-called “immersion”, meant as the extent to which senses are engaged by the mediated environment during virtual reality applications. However, it can be challenging to meet this requirement using conventional robotic design approaches that rely on rigid mechanical systems with limited workspace and bandwidth. An alternative solution can be seen in the adoption of lightweight wearable systems equipped with Neuromuscular Electrical Stimulation (NMES): in fact, NMES offers a wide range of different forces and qualities of haptic feedback. In this study, we present an experimental setup able to enrich the virtual reality experience by employing NMES to create in the antagonists’ muscles the haptic sensation of being loaded. We developed a subject-specific biomechanical model that estimated elbow torque during object lifting to deliver suitable electrical muscle stimulations. We experimentally tested our system by exploring the differences between the implemented NMES-based haptic feedback (***NMES*** condition), a physical lifted object (***Physical*** condition), and a condition without haptic feedback (***Visual*** condition) in terms of kinematic response, metabolic effort, and participants’ perception of fatigue. Our results showed that both in terms of metabolic consumption and user fatigue perception, the condition with electrical stimulation and the condition with the real weight differed significantly from the condition without any load: the implemented feedback was able to faithfully reproduce interactions with objects, suggesting its possible application in different areas such as gaming, work risk assessment simulation, and education.

## 1. Introduction

Dealing with “*haptics*” means providing *cutaneous* (tactile) and *kinesthetic* (force) feedback, two different but complementary aspects of a single and complex afferent message to our nervous system [1]. Haptic illusion is the most common approach adopted to merge virtual and augmented realities [2]: it can be achieved through vibrotactile [3] or ultrasonic [4] stimulations or with robotic force fields [5]. Depending on the desired feedback to provide the user during a virtual experience, it can be possible to adopt different technologies. Vibrotactile devices can deliver additional tactile feedback and improve, for example, human motor learning [6] or immersive virtual environments [7]. Commonly, such tools are composed of wearable vibration units or motors that can be placed on different body locations and controlled independently to generate the desired feedback [8]. Another approach to producing tactile feedback is using ultrasonic stimulation: with such methodology, it is possible to obtain acoustic radiation force, producing small skin deformations and thus elicit the sensation of touch [9]. In both cases, the limitation of tactile feedback alone during a virtual experience is of course the lack of information regarding the inertia of the object being manipulated in the scenario.

On the other side, to provide the user kinaesthetic feedback, it is necessary to generate, for example, force fields that often involve bulky devices, in which the haptic feedback is bound to limited workspaces and bandwidth [10], especially for what concerns teleoperation in both industrial [11] and surgical [12] realms.

The recent introduction of soft robotic suits, with their lightweight and higher ergonomics, motivates many researchers to develop haptic interfaces by considering such a technology [13]. Nevertheless, these garments adopt sensors and actuators able to develop advanced human-machine interfaces [14,15] with the human-in-the-loop. The inclusion of the wearer in the real-time control framework achieves a bidirectional sharing of information with the device that can be used for haptic feedback. Suppose during a specific task we tap into the central nervous system by stimulating the antagonist muscle or group of muscles. In that case, it is possible to create a more realistic haptic illusion than what the traditional techniques can provide: such an approach overcomes the bandwidth above and workspace limitations, exploiting the object interaction feeling. Recently, Neuromuscular Electrical Stimulation (NMES) has been adopted for this purpose [16,17,18] by using cues above the sensory thresholds of skin receptors. The application scenario embraces hand prostheses [19,20], remote environments during teleoperations [21], as well as for somatosensory training in post-stroke patients [22]. However, involving NMES to induce forces and movements has been less explored. Pfeiffer et al. [17] proposed a pedestrian navigation system based on NMES, in which users did not need to focus on the orientation task since a signal capable of moving the sartorius muscle was sent to change the walking direction. Kruijff et al. [18] performed an initial user test of NMES for haptic feedback, showing its potential in wearable applications. The authors also highlighted the importance of properly calibrating the human-in-the-loop system needed to stimulate the muscles to the proper extent while ensuring a high level of comfort. This point has also been evidenced by Harris et al. [23], who developed an elbow platform to enhance haptic sensations under several virtual wall hit scenarios.

The receptors’ stimulation with a proper modulation of different cues during the external object interaction makes NMES a potential tool for increasing embodiment, immersion, and ecological validity in virtual reality applications [24,25]. In addition, more complex applications (e.g., simulations for training soldiers, first-aid responders, firefighters, sports players, and rehabilitative trials) require physical embodiment through a motion tracking system and a graphical representation of the user’s body. Most rely on a realistic virtual user representation through a human avatar replicating the user’s movements [26,27,28].

In this light, we want to develop a setup to enhance the virtual scenario immersion by employing a full-body haptic system based on NMES in conjunction with a 3D visor. Notably, in this first stage, we focused on the elbow joint to develop a biomechanical model able to provide haptic illusion during virtual object interactions.

To this extent, our study was threefold: (i) developing an NMES-driven subject-specific haptic interface; (ii) evaluating it in a completely immersive virtual environment by monitoring participants’ physiological and kinematic metrics; (iii) simulating the effort of a virtual load on the users’ arm and comparing it with a real-load and a no-load condition.

## 2. Haptic Force Feedback via Functional Electrical Stimulation for Virtual Reality

### 2.1. Experimental Setup and Task

Our experimental setup (Figure 1a) involved the commercial NMES-based suit, named Teslasuit^®^ (VR Electronics Ltd., London, UK), and the head-mounted display Oculus Rift S (Facebook Technologies & Lenovo, Cambridge, MA, USA). The Teslasuit is a wearable device that incorporates ten inertial measurement units (IMUs) and 80 wireless channels for muscular electrical stimulation controlled via Wi-Fi. The virtual scenario generated using Unity 3D (Unity Software Inc., Copenhagen, Denmark, version 2019.2.13), consisted of a room in which the user (represented with a black avatar) was standing in the center while holding in the right hand a small cube. In front of the user was shown a white phantom whose arm posture the user had to match (Figure 1a). The phantom was seen by the user during the entire experimental duration (Figure 1a).

The experimenter helped the subjects wear the suit by ensuring a proper electrode positioning: this procedure was required to be started at least 20 min before the task to obtain the right fitting between the suit electrodes and the skin.

During this time frame, users were set with the metabolic consumption system.

Before the measurement, this device was warmed up for 30 min and calibrated through a high-quality calibration gas. Lastly, users placed the visor over their eyes to clearly see the virtual scenario (Figure 1a).

The task consisted of tracking, with the right arm, the phantom’s arm movement (Figure 1). The movement involved both elbow extensions (fully arm extension) and flexions (90 deg elbow angle) with a constant speed of 45 deg/s. The experimental session comprised three main conditions randomly proposed among participants:(1)Visual and Physical weight handled (0.5 kg) (***Physical***): the user received visual feedback from the virtual scenario combined with the haptic feedback of the handled physical weight;(2)Visual and NMES haptic feedback (***NMES***): the user received visual feedback from the virtual scenario combined with the haptic feedback provided by the NMES;(3)Visual feedback only (***Visual***): the user received only visual feedback from the virtual scenario without any haptic feedback.

Each condition lasted 4 min, in which a total of 32 movements (flexion and extension) were proposed. Between conditions, participants rested for 15 min in order to avoid fatigue effects. The overall session was completed in about 1 h 30 min.

### 2.2. Subjects

A group of twelve healthy, young, and right-handed participants (10 females, 2 males, 27.4 ± 3.8 years old, mean ± std, weight 62.25 ± 7.9 kg, height 165.2 ± 6.2 cm) took part in the model validations and tests. All participants provided their informed consent before the experiment, and the experimental protocol was approved by Heidelberg University Institutional Review Board (S-287/2020): the study was conducted following the ethical standards of the 2013 Declaration of Helsinki. Experiments were carried out at the Aries Lab (Assistive Robotics and Interactive Exosuits) of Heidelberg University. Subjects did not have any evidence or known history of neurological diseases and exhibited a normal joint range of motion and muscle strength.

### 2.3. NMES Calibration and Biomechanical Model

We designed a model-based real-time controller to provide NMES haptic feedback during object interaction. It consisted of the *NMES stimulation module*, developed in Unity engine^®^, which combined, in real-time, the arm kinematics to compute the respective NMES power to be delivered to the biceps or triceps muscle depending on the movement phase (i.e., extension or flexion, respectively), Figure 1b.

Our application aimed to make the virtual reality experience as immersive as possible, allowing the user to feel the weight and resistance of the visualized object during its holding and lifting. Since the heavier the actual object is, the stronger the counterforce produced on a human system, the administered artificial NMES haptic feedback has been fashioned to ensure such a sensation when a virtual object is manipulated. A prerequisite for implementing the physicality of the desired handled item was the parameterization of the same item by defining its shape (*cubic*), mass (*m_cube_*), and size (*l_cube_*). Then, it was possible to implement a biomechanical model that can modulate over time and, according to the arm’s position, the NMES acting on the user’s antagonist muscle (triceps or biceps depending on the lifting phase).

When the arm lifts an object, the most part of the work is performed by the major elbow flexion muscle (i.e., the long head of the biceps), which provides haptic feedback to the human body through the muscle spindle receptors. To achieve the same sensation in a virtual environment, the system had to stimulate its major antagonist muscle (i.e., the long head of the triceps) in order to provide the torque at the elbow level corresponding to a similar lifting task. Following the aforementioned rationale, a complementary situation occurs when the arm brings the object to the starting position; the gravity effort generates an extension torque to the elbow, which is stabilized by the triceps: to perceive it, the biceps muscle has to be stimulated (Figure 1b). The expected result is to reproduce a realistic haptic experience in the virtual world.

Before starting our experiment, we characterized the muscular response of both the biceps and triceps to different NMES stimulations in terms of the resulting measured forces. This procedure was not subject-specific: we enrolled a single sample subject to tune the parameters. We built a single-degree-of-freedom elbow platform to calibrate the NMES feedback, as shown in Figure 2. During the controlled NMES muscle contraction, the force sensor measured the respective end-effector force ($F^{stim})$ generated by the biceps/triceps stimulations (Figure 2).

The calibration setup consisted of horizontal arm support at the subject’s shoulder height, resulting in an elbow angle *q* equal to 45°, and a customized force sensing system holder positioned to match the subject’s wrist anatomical landmark (PL) where the force output was measured. A force sensor (Futek, FSH04416, Irvine, CA, USA) has been mounted in the force-sensing system to record and transmit data to a dedicated acquisition board (Quanser QPIDe, Markham, ON, Canada) at 1 kHz.

During the calibration, we administered to the subject muscle (biceps/triceps) ten increasing NMES stimulations with a duration of 2 s each, followed by a 5 min rest phase.

Two distinct acquisitions were performed to the right triceps and the biceps muscles.

We modulated the NMES parameter Pulse Width, *PW* (half-wave width range between 1–60 μs, normalized in percentage with an interval of 10% between each stimulus) during each stimulation and saved the respective force output read by the load cell. The stimulation frequency was fixed at 60 Hz, while the maximum current per channel was equal to 150 mA and the maximum possible voltage was 60 V. We obtained the desired relationship between the administered pulse width *PW* and the corresponding output force recorded through the force sensor, $F^{stim}$, generated against the flat and rigid force-sensing system (Figure 3), with an accuracy equal to *R*^2^ = 0.9834:(1)$$F^{stim}=a\cdot PW^{2}-b\cdot PW+c$$
where *a*, *b*, *c* are constants, that, in our subject-specific case, assumed values equal to 0.0028, 0.1123, and 0.5816, respectively. This force acted on the elbow joint by following the relationship:(2)$${\vec{\tau }}_{elbow}=F^{stim}\times r_{m}$$
where $r_{m}$ is the force’s moment arm.

In order to provide haptic feedback during the experiment, we modulated the net torque at the elbow level using muscle stimulations. During free motions, the joint torque can be modelled as:(3)$${\vec{\tau }}_{elbow}={\vec{\tau }}_{arm}+{\vec{\tau }}_{object}$$
where ${\vec{\tau }}_{arm}$ is the biomechanical torque of the forearm acting on the joint during movements, while ${\vec{\tau }}_{object}$ is the contribution of the simulated virtual interaction. Assuming the arm is parallel to the chest (i.e., shoulder angles = [0 0 0]), we can model ${\vec{\tau }}_{object}$ as:(4)$${\vec{\tau }}_{object}=(I_{object}+m_{object}\cdot r_{d}{}^{2})\cdot \ddot{q}+m_{object}\cdot g\cdot r_{d}$$
where *q* is the elbow angle acquired from the NEMS system IMUs [29], $I_{object}$ and $m_{object}$ are, respectively, the moment of Inertia and the mass of the object (of which it is desired to simulate the holding during the task), and $r_{d}$ is the distance between the object’s barycenter and the elbow joint fulcrum.

To provide participants with the tuned haptic feedback (*PW*) according to the elbow kinematics (*q*) and the object, the following system has to be solved:(5)$$\{\begin{array}{c} {\vec{\tau }}_{object}+{\vec{\tau }}_{arm}=\left(a\cdot PW^{2}-b\cdot PW+c\right)\times r_{m}\\ {\vec{\tau }}_{object}=(I_{object}+m_{object}\cdot r_{d}{}^{2})\cdot \ddot{q}+m_{object}\cdot g\cdot r_{d} \end{array}$$
where ${\vec{\tau }}_{arm}$ is the torque provided by the musculoskeletal system. By solving the above system, the Pulse Width modulation was tuned in order to generate a resistive action on the elbow, considering the inertial properties of the object as:(6)$$PW=\frac{{\vec{\tau }}_{arm}+b\pm \;\sqrt{b^{2}-4\cdot a\cdot \left[c-\left((I_{object}+m_{object}\cdot r_{d}{}^{2}\right)\cdot \ddot{q}+m_{object}\cdot g\cdot r_{d}\right]}}{L_{arm}\cdot sin\;\left(q\right)}$$
where $r_{m}=L_{arm}\cdot sin\;\left(q\right)$, and $L_{arm}$ is equal to the subject’s forearm length.

As the second step of the calibration, we performed a brief and ad hoc subject safety procedure before starting the experiment to set the NMES intensity’s minimum and maximum values. Since the skin impedance is vastly different among subjects, this step was mandatory before the suit utilization and was crucial to avoid uncomfortable events.

### 2.4. Outcome Measures

To assess the human performance, we quantitatively highlighted the onset of fatigue by measuring the metabolic expenditure with a wearable system (K5, Cosmed), known for being reliable during several exercise modalities [30,31,32,33].

To evaluate the metabolic consumption variations occurring in the three experimental conditions, we evaluated the *Respiratory Exchange Ratio* (*RER*) [34,35], from the ergospirometry variables provided by the COSMED K5, which was operating in mixing chamber mode. Specifically, the volume of oxygen consumption (VO2) and carbon dioxide production (VCO2) were assessed for computing the *RER* as follows:(7)$$RER=\frac{VCO2}{VO2}$$

*RER* values are typically comprised between 0.7 and 1.2. During non-steady-state and high-intensity exercises, the volume of the carbon dioxide produced by the human body increases due to hyperventilation with a consequent rise of the *RER*.

From the NMES system IMUs, we recorded elbow angle trajectories at 100 Hz and offline filtered using a 6th order low-pass Butterworth filter with a 10 Hz cutoff frequency. We extrapolated the indicators for characterizing subjects’ kinematic performance as the primary output.

The *Absolute Error*, *AE**_R.O.M._*** (deg), which analyses performance accuracy during the tracking task, is computed by subtracting the ideal R.O.M. completed by the *phantom* from the R.O.M. made by the subject:(8)$$AE_{R.O.M.}=abs(R.O.M._{phantom}-R.O.M._{user})$$

The *Root Mean Squared Error* (*RMSE*) measures the participant’s elbow angle trajectory deviation from the ideal *phantom* trajectory. It is defined as:(9)$$RMSE=\sqrt{\frac{1}{N}\sum_{i=1}^{N}{\left(q_{phantom}-q_{user}\right)}^{2}}$$
where $q_{user}$ is the *user* elbow angle trajectory, $q_{phantom}$ is the *phantom* elbow angle trajectory, both evaluated at sample *i*, and *N* is the total number of samples considered on the entire trial.

We evaluated the fitting between the ideal trajectory of $q_{phantom}$ and the user trajectory $q_{user}$ using the correlation coefficient *r*^2^.

Moreover, we considered the *Normalized Smoothness*, following the approach of Balasubramanian et al. [36], which is a slightly modified version of the original *Spectral Arc Length* (*SAL*) definition:(10)$$SAL≜-\int_{0}^{\omega_{c}}{\left[{\left(\frac{1}{\omega_{c}}\right)}^{2}+{\left(\frac{d\hat{V}\left(\omega \right)}{d\omega }\right)}^{2}\;\right]}^{\frac{1}{2}}\;d\omega ;\;\hat{V}\left(\omega \right)=\frac{V\left(\omega \right)}{V\left(0\right)}$$
where *V*(*ω*) is the Fourier magnitude spectrum *v*(*t*), $\hat{V}\left(\omega \right)$ is the normalized magnitude spectrum, normalized with respect to the DC magnitude *V*(0), and *ω_c_* is fixed to be 40*π* (corresponding to 20 Hz). In this modified version, we adopted the SPARC for **SP**ectral **ARC** length by setting: (11)$$\omega_{c}≜\;min\;\left\{\;\omega_{c}^{max},\;min\;\left\{\omega ,\;\hat{V}\left(r\right)\;\langle \;\overset{=}{V}\;\forall \;r\rangle \omega \;\right\}\right\}$$

We evaluated, for ***NMES*** and ***Physical*** conditions, the torque at the elbow generated by virtual and real weight, respectively.

Finally, participants answered on a 7-point Likert scale (from −3 = completely disagree, to +3 = fully agree) to evaluate the Pleasantness and Naturalness of the three different experimental conditions [37]. This test was essential to understand the ecological validity of the immersive environment.

The metrics *AE_R.O.M._*, *RMSE*, *Normalized Smoothness*, *RER*, were averaged over time.

### 2.5. Statistical Analysis

We used a repeated-measures analysis of variance (rANOVA) on the dependent variables, and we considered as the within-subjects factor (“*Feedback*”) the kind of provided haptic feedback (***Physical***, ***NMES***, ***Visual***). Data normality was evaluated using the Shapiro–Wilk Test, and the sphericity condition was assessed using the Mauchly test. Statistical significance was considered for *p*-values lower than 0.05. Post hoc analysis on significant main effects was performed using Bonferroni corrected paired *t*-tests (*p* < 0.0025).

For the Likert scale outcomes, *Pleasantness* and *Naturalness*, non-parametric paired tests were employed. The Kruskal–Wallis test was used for comparisons among the three trials (*p* < 0.05), while the Wilcoxon signed-rank test was used for the paired comparisons (*p* < 0.0025). Outliers were removed before any further analysis using a Thompson Tau test.

## 3. Results

### 3.1. The NMES Feedback Is Comparable to the Physical in Terms of Torque

Figure 4a depicts the torque comparison between the torque obtained with the ***NMES*** condition *(*${\vec{\tau }}_{elbow}$) and the one obtained during the ***Physical*** condition (${\vec{\tau }}_{object}$) for a representative subject. From this comparison, we found high *r*^2^ values for all subjects (mean ± SE: 0.993± 0.002) and low differences by means of *RMSE* values (mean ± SE: 0.116 ± 0.020 (Nm)), Figure 4b. This result validates our calibration, and it evidences the appropriateness of our approach for all participants.

### 3.2. NMES Condition Does Not Influence the Kinematic Accuracy

Figure 5 displays the elbow joint kinematics parameters computed across all subjects during the task and among the ***Visual***, ***Physical***, and ***NMES*** conditions. Through the first three parameters (*AE_R.O.M_***_._**, *RMSE*, and *r*^2^), we evaluated the accuracy in faithfully reproducing the given trajectory.

We encountered similar performances among the three proposed conditions, highlighting that the NMES-based haptic feedback (***NMES*** condition) does not interfere with the physiological range of motion. The statistical analysis confirmed such a result: for the *AE_R.O.M._* (Figure 5a), we found no significant effect between the three conditions (‘*Feedback’* effect: F = 0.035, *p* = 0.966). We also reported the *RMSE* (Figure 5b) and *r*^2^ (Figure 5c) with analogous findings for both the parameters (‘*Feedback’* effect: F = 0.151, *p* = 0.861 and F = 0.300, *p* = 0.744, respectively). Moreover, we analyzed the *Normalized Smoothness* of participants’ movements compared to the reference trajectory. As expected, we found that the proposed NMES-based haptic feedback, due to the delivered muscle stimulation, partially affects the smoothness of the natural movement. This downside of our feedback was confirmed by the statistical analysis. The rANOVA evidenced a significant effect of the feedback (‘*Feedback’* effect: F = 5.523, *p* = 0.013). The subsequent post hoc analysis showed a significant difference between the ***Physical*** and ***NMES*** conditions (*p* = 0.0082). The other two comparisons denoted no significant differences (***Visual-Physical*** *p* = 0.2727, ***Visual-NMES***: *p* = 0.05).

### 3.3. Metabolic Consumption during the NMES Condition Is Comparable with the Physical One

We evaluated the metabolic consumption via the *Respiratory Exchange Ratio* (*RER*) parameter to understand if the exercise intensity changed during the three experimental conditions. The results are illustrated in Figure 6, which shows, as expected, that the lower intensity of the exercise was obtained during the ***Visual*** condition. From the statistical analysis with rANOVA, we highlighted an effect of the condition (‘*Feedback’* effect: F = 18.226, *p* < 0.001). From further post hoc analysis, we found a significant difference between the conditions ***Visual*** and ***Physical*** (post hoc: *p* = 0.001) and between the conditions ***Visual*** and ***NMES*** (post hoc: *p* < 0.001). A noteworthy result is the non-significant one obtained between the ***Physical*** and ***NMES*** conditions, which highlights the similarity in fatigue between the physical object handled and the NMES-based artificial stimulus.

### 3.4. Naturalness and Pleasantness

The *Naturalness* of the experiment was significantly higher in the conditions ***NMES*** and ***Physical*** than in the ***Visual*** condition, as is shown in Figure 7. The statistical analysis with Kruskal–Wallis tests confirmed this result, highlighting a significant effect depending on the feedback (‘*Feedback*’ effect: χ^2^(2) = 12.193, *p* = 0.002). The following Wilcoxon signed-rank test showed that the sensation with the ***NMSE*** condition was perceived to be more natural than the one with the ***Visual*** feedback (Z = −2.264, *p* = 0.024). On the contrary, no significant differences were detected between the task during the ***NMES*** condition and the one during the ***Physical*** condition (Z = −1.633, *p* = 0.102), highlighting the faithfulness of the proposed feedback with stimulation compared to the natural sensation. As expected, we found significant differences between the ***Physical*** and the ***Visual*** condition (Z = −2.262, *p* = 0.023). Regarding the *Pleasantness*, users perceived the NMES-based haptic feedback (***NMES*** condition) to be slightly uncomfortable, as shown in Figure 7. However, no significative feedback effect was detected (‘*Feedback*’ effect: χ^2^(2) = 0.892, *p* = 0.640).

## 4. Discussion

Virtual reality (VR) and augmented reality (AR) are two forms of modern technological advancements that have revolutionized the standard concept of visual communication over the years. However, despite their broad expansion, there is still a wide gap in their practical applications (e.g., emergency simulations, teaching, surgical training) due to the lack of immersive interactions that can be assimilated into tangible experiences. The missing piece is to interact with virtual objects that can be perceived as authentic by the human body.

### 4.1. NMES Feedback Reliability and Its Quantitative Assessment

The proposed study revealed the feasibility of a multimodal technological system combining Neuromuscular Electrical Stimulation (NMES) provided using a wearable suit with VR in order to increase the immersive sensation of a weightlifting task within a virtual environment. Based on the concept that the feeling of lifting an object could be obtained by providing electrical stimulation to the antagonist’s muscles to those exerting the movement, we developed a biomechanical model able to give a sensory response based on the real-time user’s elbow movements. The results from 12 volunteers provided experimental evidence that the NMES-based haptic feedback robustly simulates the physical exertion of a real object. Such a finding was possible thanks to a priori calibration which allowed a robust biomechanical model suitable for all the participants to be obtained. As highlighted by an early study with NMES for haptic feedback [18], the calibration phase is crucial to properly stimulate the muscle, detect noticeable pose changes, and enhance user comfort. In their study, Kruijff et al. [18] showed the importance of a proper calibration to perceive the right amount of current without generating user discomfort. For this reason, we performed an isometric calibration process before the experiments. This preliminary procedure is one of the most delicate steps that for traditional systems with electrodes requires the accurate positioning of them, a factor that was greatly simplified by the use of our wearable device; in fact, the latter allowed us to obtain a biomechanical model suitable for different subjects with slightly different anthropometric characteristics.

The study’s central finding is related to the *kinematic* reliability of the simulated weight and a comparable *metabolic* consumption between ***Physical*** and ***NMES*** conditions. These results are consistent with studies found in the literature, highlighting that NMES is a well-suited technology for providing more realistic haptic feedback during interaction with objects in a virtual environment [16]. Lopes et al. [24,38] explored how to integrate haptics to walls and heavy objects in VR through NMES: they showed how adding haptic feedback through electrodes on the user’s arms could increase the sense of presence in the virtual interactive application. However, no quantitative analysis of system performance was carried out. In the current study instead, two of the main subjects’ physiological metrics have been analyzed: kinematic performance and metabolic consumption.

First, the recorded kinematic measurements related to the accuracy of the movement (*AE_R.O.M._*, *RMSE*, and *r*^2^) showed that haptic feedback via the ***NMES*** condition did not affect the final kinematics, rendering the movement as accurate as in conditions without haptic feedback (***Visual***) or with the real weight (***Physical***).

On the other side, the metabolic consumption outcome (*RER*) revealed that NMES-based haptic feedback (***NMES***) was assimilable to the ***Physical*** condition, and in both cases, as hypothesized, the metabolic consumption was higher compared to the condition without haptic feedback (***Visual***). This result is consistent with previous works, which showed that the *RER* increase with the exercise intensity [34,35]. The sensation of muscle activation generated by the ***NMSE*** condition was comparable to that required during the ***Physical*** condition yielding similar metabolic demands. Finally, we recorded users’ opinions from the questionnaire (7-point Likert scale), which revealed that the *Naturalness* was significantly higher during the ***NMES*** and ***Physical*** conditions compared to the condition without haptic feedback (***Visual***).

### 4.2. Integration of NMES-Based-Haptic Feedback in Virtual Scenarios

The previous findings highlight the potential of the implemented NMES-based haptic feedback in multiple application areas. Interaction with virtual objects of different nature, capable of returning not only visual feedback but also haptic sensations, would increase the chances of learning more complex tasks [39,40,41]. In fact, to perceive the external environment, our brain uses multiple sources of sensory information derived from different modalities, and vision is only one of the several systems involved in the sensory process. A stimulation capable of being assimilated with an actual physical condition and integrating the various perceptive information is an essential step in granting cognitive benefits, such as an increased embodiment and involvement in the virtual scenario [37,42,43]. Our interface represents the first step in developing a virtual environment fully parameterizable and modellable according to the main characteristics of the objects to be manipulated and usable in the field of simulation, such as industrial safety and surgical training.

### 4.3. Limitations

Our system is still embryonic: firstly, more muscles would be necessary to appreciate the ***NMES*** haptic feedback entirely. Even if participants appreciated the feedback and considered it as natural as a real weight, they complained about the lack of stimulation from other muscle channels (e.g., shoulder deltoid muscles and forearm muscles). This step would require a more complex biomechanical model, for which it will be necessary, in the future, to include a preliminary electromyographic study, or a simulative environment, depending on the desired movement.

Secondly, more degrees of freedom should be included in the virtual scenario: since the adopted suit is able to provide full-body stimulation, it would be interesting to study more complex movements involving a more significant number of degrees of freedom. All these improvements would benefit even the so-called “engagement,” an aspect widely considered in the field of pure AR/VR research, which will certainly be included in our future studies.

Another aspect that affected the *Pleasantness* of the task was the lack of receiving feedback on the hand palm during the ***NMES*** condition, where, in the virtual scenario, the object was displayed. To validate our model, we decided to place the virtual object directly on the palm so as not to introduce collisions, which would have required additional computation. However, this is something we will improve in the future by including a vibrotactile surface in order to provide tactile sensation (e.g., vibrotactile gloves).

In addition, our ***NMES*** haptic feedback affected the movement smoothness with respect to movement with the physical weight. This physiological effect generated by electrical stimulation on afferent pathways can be reduced by implementing an improved stimulation paradigm. Only the comparison with the ***Visual*** condition highlighted this effect in our data, thus making this aspect of no concern.

Moreover, the developed haptic feedback was tested only on a few healthy subjects to probe the system’s feasibility. The availability of a single suit size precluded the inclusion of a wide range of participants in terms of anthropometric measures. This aspect has also affected the results that emerged from the statistical analysis. In the future, further subjects should be added to increase the sample size and the reliability of the results.

In addition, the evaluation of the metabolic cost significantly contributed to the feedback assessment, bringing with it quantitative evidence that the ***Physical*** and ***NMES*** conditions were comparable. However, this evaluation system affected both the duration and the task ergonomics: in the future, this measurement will be evaluated at the discretion of the users who will have to use the interface.

## 5. Conclusions

The current study presents a novel paradigm to provide haptic feedback via neuromuscular electrical stimulation that can increase the immersion and the quality of the experience during the execution of a task in a virtual reality environment.

Our results, on a small sample of healthy subjects, showed the potential of an NMES-based haptic interface and highlighted, for the first time, the quantitative fatigue consumption with respect to a comparable physical condition.

The real-time biomechanical model ran during the task execution represents a starting point for fully customizable haptic feedback. The employment of the current system, with proper modifications (e.g., the use of multiple suits of different sizes or single electrode systems), could be used for a wide range of applications involving the entire upper body that can be from surgical training to rehabilitation.

## Figures and Tables

**Figure 1 sensors-22-05069-f001:**
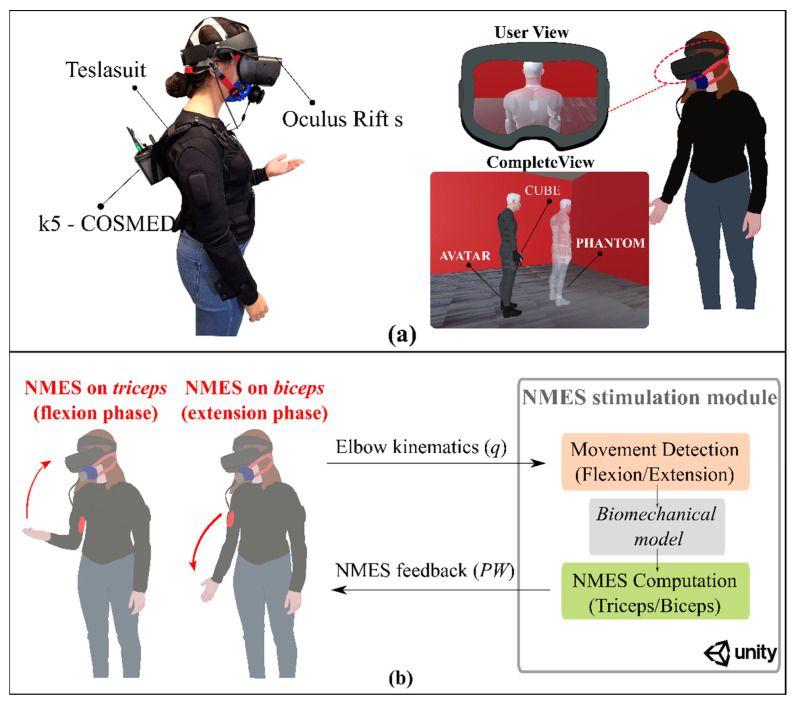
(**a**) Experimental setup: the subject while wearing the NMES-based suit (Teslasuit), the 3D visor (Oculus Rift s), and the metabolic consumption device (k5-Cosmed). On the right is shown the scenario rendered on the 3D visor during the task (user view). Underneath represents the complete view of the implemented virtual scenario in which the user can see its posture (black avatar) and the one to match (white avatar) while handling the virtual cube (cube). (**b**) Real-time control scheme of the NMES-based haptic feedback: the biomechanical model implemented within the *NMES stimulation module*, received as input of the elbow angle read by the suit sensors and, depending on the phase of the elbow movement (flexion/extension, red arrows), delivered electrical stimulation to the respective muscle antagonistic to the one activated during the detected phase (triceps/biceps, red areas).

**Figure 2 sensors-22-05069-f002:**
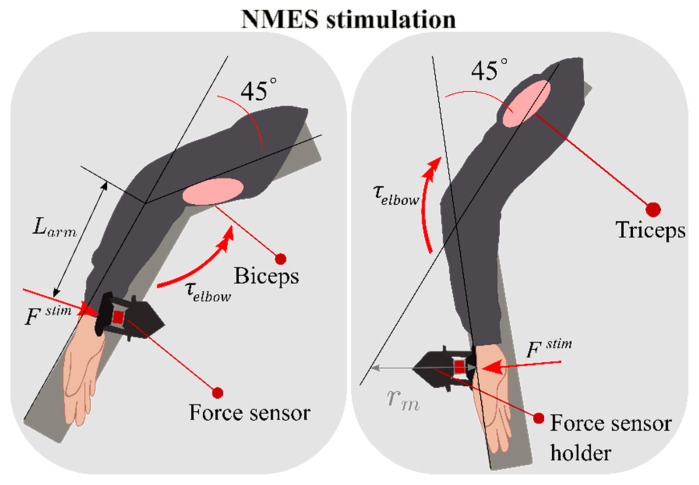
Calibration setup: top-view of the single-degree-of-freedom elbow platform to calibrate the NMES system. The whole arm was lying on the support; the wrist was positioned in concomitance with the force sensor holder, against which the subject applied the force generated after the NMES stimulation. On the left panel, the NMES stimulation targeted the biceps muscle (pink-colored oval), the resultant force generated (*F^stim^*), and the torque acting on the elbow (${\vec{\tau }}_{elbow}$). On the right panel, an equal representation of when the NMES stimulation targeted the triceps muscle (pink-colored oval).

**Figure 3 sensors-22-05069-f003:**
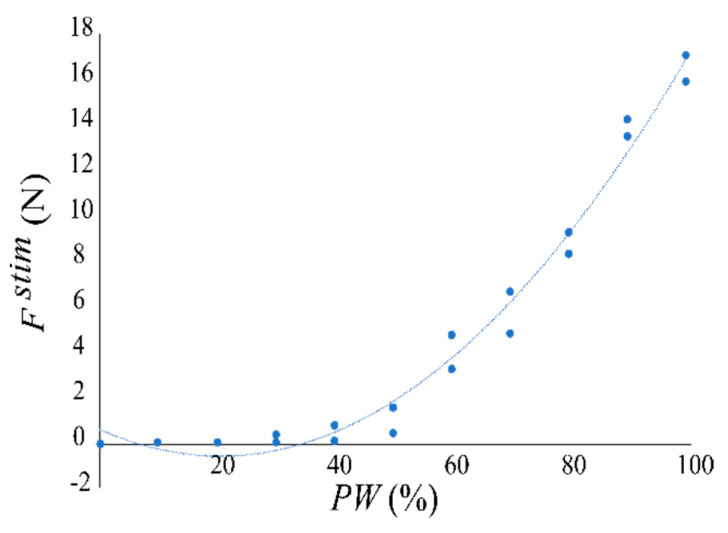
Interpolation of the calibration results relative to the stimulation provided on the *biceps* muscle for a sample subject. On the *x*-axis, the *PW* values given to the subject via the NMES system are represented. On the *y*-axis, the muscle response with respect to the force measured by the force sensor is depicted. *PW* range is between 1 and 60 μs, normalized in percentage with an interval of 10% between each delivered stimulus.

**Figure 4 sensors-22-05069-f004:**
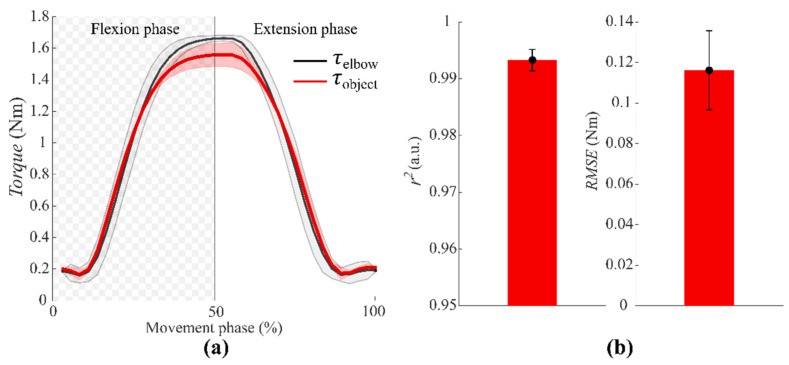
(**a**) Torque profile comparison (mean ± SE for a sample subject) between the torque obtained with the ***NMES*** condition (${\vec{\tau }}_{elbow}$) and the one obtained with *the **Physical*** condition (${\vec{\tau }}_{object}$ ). (**b**) *r*^2^ (left) and *RMSE* (right) metrics to evaluate the accuracy of the torque comparison.

**Figure 5 sensors-22-05069-f005:**
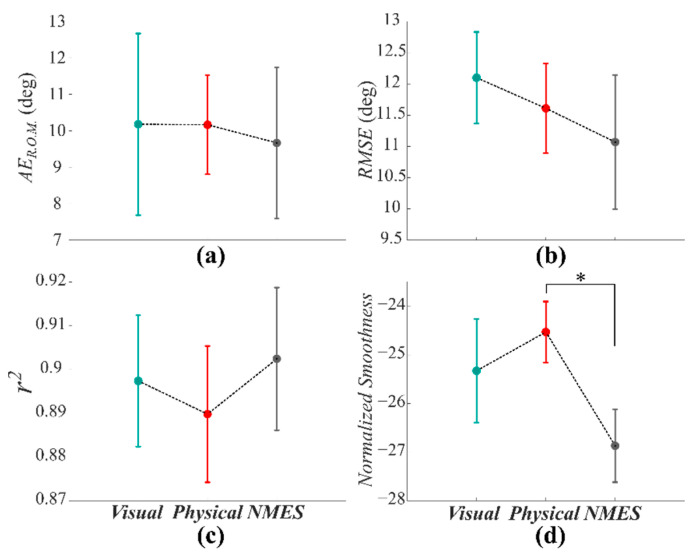
*Kinematic parameters* were computed among subjects for the three feedback conditions (***Visual***, ***Physical***, and ***NMES***). (**a**) the *AE_R.O.M._*, (**b**) the *RMSE*, (**c**) the *r*^2^, and (**d**) the *Normalized Smoothness*. Significant differences (*p* < 0.0025) are marked with an asterisk.

**Figure 6 sensors-22-05069-f006:**
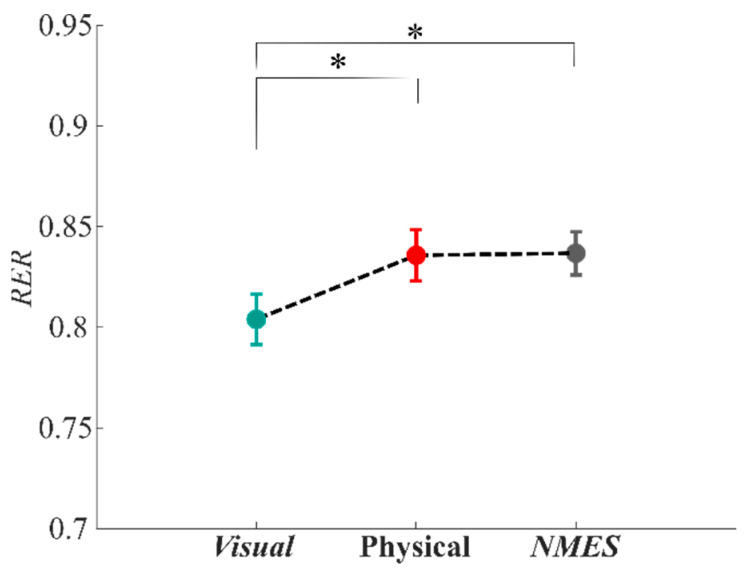
Metabolic cost result: the *Respiratory Exchange Ratio* (*RER*) parameter. Results are reported as mean and standard error values for the three feedback conditions (***Visual***, ***Physical***, and ***NMES***). Significant differences (*p* < 0.0025) are marked with an asterisk.

**Figure 7 sensors-22-05069-f007:**
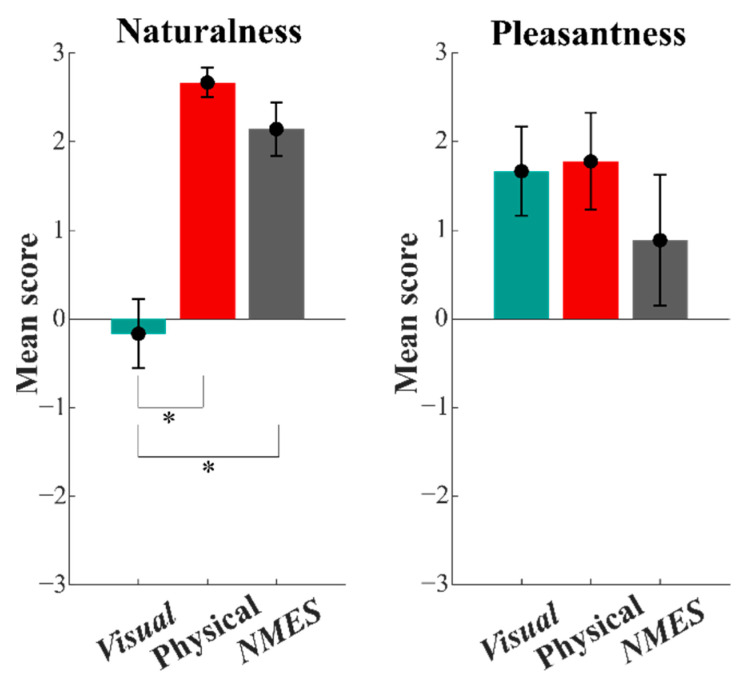
Results for *Pleasantness* and *Naturalness* outcomes from the Likert scale. Results are reported as mean and standard error values for the three feedback conditions (***Visual***, ***Physical***, and ***NMES***). Significant differences (*p* < 0.0025) are marked with an asterisk.

## Data Availability

The datasets generated and/or analyzed for this study are available from the corresponding author on reasonable request.

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
