# Peer review of "Rendering Immersive Haptic Force Feedback via Neuromuscular Electrical Stimulation"

_sensors, 2022, doi:10.3390/s22145069_

Round 1

Reviewer 1 Report

[General Comments]

The authors used the output of the antagonist muscle from functional electrical stimulation as the reaction force from the VR object to represent the joint torque required from the VR object and body model. A calibration method was developed to match the magnitude of the electrical stimulation with the magnitude of the force generated by the muscles. This enabled to achieve approximately the desired joint torque. The manuscript can be evaluated as a starting point for a fully customizable haptic environment. 

The presentation of the weight sensation of the grasped object to the elbow joint using FES has been reported for a long time and is not highly novel, but the quantitative evaluation focusing on the degree of motion similarity seems new contribution. However, it is necessary to separately discuss to what extent it contributes to the sense of force, the accuracy of the presentation of VR objects, and the immersive feeling of the user. Moreover, since the movements performed in the validation experiment were single, simple ones, it is difficult to say that the effect has been adequately verified. 

In addition, there are several unclear points, including the following.

[Specific Comments]

1: In the section 2.1, the procedure of experients is unclear. Are "Real weight handled (0.5 kg)(in L112)" and "holding in the right hand a small cube(L84)" the same? 

In the "same" case, the subject also has something during (H), but as Figure 1 shows it doesn't. In the "different" case, the subject has a virtual cube during (H). In the "different" case, (H) has a virtual cube; Comparing (R) and (H), did holding an object or not have any effect on the accuracy of the force or the strength of the motion? I understand the authors mentioned about Pleasantness in L437-442. 

2: In the section 2.3, the approximate expression in Fig. 3 is equation (1), but is it OK to have a negative part? Should the approximation equation be square? How about exponential?

3: Ditto: L191-L230, these show a simple motion of bending the elbow, but it is hard to understand. confusion. Please add a picture showing geometric conditions of Fs, r_m, r_d, L_arm, etc.

4: In the section 2.4, the time variation and RMS using the error with the reference and the correlation coefficient with the reference seem to be essentially the same when averaged over time. Figure 5 shows only a similar trend, although this is partly due to the large variation and the lack of significant differences. 

5: In the section 3, it should be indicated in section 2.4 that torque is also evaluated.

6: Ditto: in Figure 4, the torques of the stimulus and control groups appear to be well matched in the overall profile, but the maximal value seems to differ by about 10%. Is that OK?

7: In Discussion: As the authors mentioned, it is difficult to strongly guarantee the results because the movements are simple and one-patterned. At the very least, I would like to see a comparison of the results for different velocity patterns or different angle patterns.

Reviewer 2 Report

This paper treats an interesting research of neuromuscular electrical stimulation for haptic feedback. The research method is well described and reliable. I think the technology is an early stage so this paper has useful information to researchers.

There are weak points of this research description to consider the current art of this technology. So, I request author adds more description to the two questions;

(1)  The sample size is too small and they only evaluate their results based on the significant difference without considering effect size. Could you please add this weak point on the discussion part. In addition, if you find overestimation of your results, then revise such parts in the whole part of the paper.

(2)  The title includes “Enhancing Immersive” experience, but you only ask “Naturalness” and “Pleasantness”. What do you think of evaluation of immersiveness in the AR/VR research? Have you checked engagement to the mediated environment?

Reviewer 3 Report

This paper presents a biomechanical model for elbow torque activities. The paper illustrate how the biomechanical model was achieved and an experiment to test the validity and subjective nature of the model.

I have the following comments to the authors:

- The paper is very hard to read. It requires an extensive proof reading and change of sentence structures. Moreover, the paper needs formatting as the formatting is inconsistent (fonts, paragraph spacing….) . Figure captions are very confusing. Figure 7 is better with colors. The language made it hard to follow the contents of the paper.

- The contribution of the paper is not highlighted. It seems the paper adopts methodologies done in previous work. The contribution should be clearly mentioned in the conclusion and/or introduction.

Reviewer 4 Report

This is interesting research that used neuromuscular stimulation on the antagonist’s muscles to create a haptic sensation of being loaded. This is the key innovation of this study but also the key confusion in this article. The point is that “haptic feedback” was intuitively referred to as a tactile (or touch) sensation and this gave confusion on how a neuromuscular stimulation produced a “ touch” sensation. Nevertheless, the innovation is much appreciated, and I strongly recommend the authors highlight this feature (I refer to the sensation of the hand being loaded by stimulating the antagonistic muscles, instead of a tactile touching haptic) in the title of the research, abstract, and any place, where applicable. Here are some other comments:

Line 15: The full body sensorized suit. The “sensorized” gives an impression of “touch feeling”. It would be appreciated if the author could repack it to highlight their strengths.

Line 18: N”E”MS-based haptic feedback, an actual lifted object, and a baseline condition without haptic. The conditions did not seem to be parallel. To me, “an actual lifted object” seemed to be equivalent to “a baseline condition without haptic”. The key condition is recommended to be clearer. The same applied to the method section.

Line 20: The conditions used in the abstract result section (condition with electrical stimulation) do not have the same condition name as described in the last sentence. It is recommended to be consistent so that readers can follow easily. The same applied throughout the manuscript.

Line 35: For the literature review, I recommend to repack the review and highlighting the differences between tactile/or vibration haptics and the neuromuscular stimulation haptics. What are their differences in applications and their respective strengths?

Line 110: Need to be more clear. E.g. Visual and real weight handled. Does it mean without VR feedback and conduct in real environment?

Line 236: I recommend to separate outcome measures and statistical analysis. It is a bit messy here. For statistical analysis, a brief summary on the outcome measures can be followed, while the outcome measures can focus on the data derivation/reduction, calculation methods.

Line 448: An explicit limitation section shall be put forward in the discussion, e.g. external validity on participant selection, internal validity on the experiment and evaluation methods.

Round 2

Reviewer 1 Report

[Genaral Comments]

The authors modified the manuscript properly according to the comments in the previous review except the followings. The manuscripts became clearer than the original, but there are several points that can be improved.

[Specific Comments]

Point 4: The authors misunderstood my comment. The comment meant the equation (1) solved F stim from PW, there F stim could be negative, but actually, F stim was always positive as shown in Fig. 3. Of course, the approximation equation does not need to be square, so it can be other form like as exponential. But why was the quadratic form chosen? I guess that the equation is used to solve PW from F stim, thus the range of PW where F stim is negative is virtually unrelated. Moreover, the authors may want to solve easily an inverse equation of F stim by quadratic form. I mentioned an example of exponential form above because it can be solved analytically. 

Point 5: Please add variables for length, rm and Larm in Fig. 2. 

Reviewer 3 Report

I thank the authors for their reply. My concerns have been answered. 

Author Response

We sincerely thank the Reviewer for the feedback. We are satisfied with the new version of the manuscript and with the elucidation of every point raised during the revision process.